# OpenReview forum: "Can LLMs Move Beyond Short Exchanges to Realistic Therapy Conversations?"
_ICLR.cc/2026/Conference — ICLR 2026 Conference Withdrawn Submission_

### Official Review · Reviewer_Qzrb · 2025-10-31

**Soundness:** 3
**Presentation:** 2
**Contribution:** 3
**Rating:** 4
**Confidence:** 3

**Summary:**

This paper introduces CareBench-CBT, a large-scale, clinically validated benchmark designed to evaluate the counseling competence of LLMs based on Cognitive Behavioral Therapy (CBT). The authors identify critical gaps in existing benchmarks, namely their reliance on synthetic or single-turn QA data and their failure to capture the formal structure of therapeutic conversations.

CareBench-CBT addresses this by unifying three complementary components:
- Knowledge-based QA: 640 items (430 public, 210 expert-rephrased "private" items) to test factual CBT knowledge and robustness against "teaching to the test."
- Case vignette classification: 60 expert-authored clinical vignettes to evaluate diagnostic and clinical reasoning abilities.
- Multi-turn counseling dialogues: 256 complete, anonymized counseling sessions (totaling 7,680 turns) annotated by 21 licensed professionals. These dialogues are aligned with the formal structure of CBT.

The authors use this benchmark to evaluate 18 state-of-the-art LLMs. Key findings show that while models perform well on public QA, their scores degrade significantly (avg. 21-point drop) on the private, rephrased set. Furthermore, nuanced clinical reasoning (vignette classification) remains a major challenge for most models, and even the best-performing LLMs fall substantially short of human counselor performance in long-form, multi-turn dialogue competence.

**Strengths:**

- The benchmark's core strength is its high-quality, clinically-grounded data. 21 licensed professionals validated all items using robust protocols (CTS-R, MITI) and achieved high inter-rater reliability (Cohen's K > 0.89). This rigor is applied to evaluating "process-level competence" using 256 complete, multi-turn dialogues aligned with formal CBT structure.

- The design of the Knowledge-based QA task is a significant contribution. By creating a "private" set of 210 expert-rephrased items and comparing performance against the 430 public items, the authors provide clear, quantitative evidence of performance degradation. The average 21-point accuracy drop effectively demonstrates that high scores on public benchmarks are often a result of memorization, not robust, generalizable clinical knowledge.

**Weaknesses:**

- **Overly Broad Title and Limited Scope**: The paper's title is misleading. The benchmark is exclusively focused on CBT. While CBT is a widely-used modality, its conversational structure is not representative of all mental health counseling (e.g., psychodynamic, humanistic, DBT). The benchmark's findings and utility are therefore limited to CBT and cannot be generalized to "realistic therapy conversations" as a whole, as the title implies.

- **Data Provenance Ambiguity**: The paper states the 256 dialogues were "provided by 21 licensed mental health counselors" and "collected and anonymized from authentic therapeutic sessions." However, the exact source and nature of these sessions are not detailed. This ambiguity makes it difficult to assess potential sampling biases (e.g., in patient demographics, severity, or cultural diversity), which is a critical detail for a benchmark claiming high clinical validity.

- **Circular Human Baseline Evaluation**: The main "Human" baseline score in the dialogue evaluation (Figure 5) is derived from the LLM-as-a-Judge ensemble scoring the ground-truth human therapist responses. This means the benchmark used to demonstrate the "gap" between LLMs and humans is itself an LLM-generated metric. This is a circular methodology. A more robust baseline would involve human experts scoring the human therapist responses to set a "gold standard" bar, rather than relying on an LLM to judge a human to set the bar for other LLMs.

**Questions:**

Can the authors provide more detail on the source of the 256 authentic counseling dialogues? Were they from a specific set of clinics, training archives, or multiple private practices? This information is crucial for understanding potential sampling biases (e.g., in patient demographics, problem severity, or cultural diversity).

---

> ### Author Response · Authors · 2025-11-18
> **Part 1**
>
> Weaknesses:
>
> > 1. Overly Broad Title and Limited Scope: The paper's title is misleading. The benchmark is exclusively focused on CBT. While CBT is a widely-used modality, its conversational structure is not representative of all mental health counseling (e.g., psychodynamic, humanistic, DBT). The benchmark's findings and utility are therefore limited to CBT and cannot be generalized to "realistic therapy conversations" as a whole, as the title implies.
>
> **Response:**
>
> Thank you for this thoughtful comment. We fully agree that our benchmark is grounded in CBT and does not attempt to cover the full diversity of therapeutic modalities (e.g., psychodynamic, humanistic, DBT). But our use of the term “realistic therapy conversations” is not intended to imply that we cover all modalities of psychotherapy. Rather, “realistic” is contrasted with existing isolated QA and single-turn benchmarks that overlook the extended, adaptive nature of real interactions. Our main contribution is to move from such short exchanges to fully multi-turn sessions that reflect authentic therapeutic structure (rapport building, guided exploration, intervention and closure). These process-level properties are shared across many evidence-based modalities, even though the specific techniques and theoretical underpinnings differ. Thus, while the content of our benchmark is CBT-based, the interactional competencies it measures (e.g., empathy, therapeutic attunement, intervention quality) can generalize to realistic therapeutic dialogue beyond CBT alone.
>
>
> In addition, we appreciate the reviewer’s observation. To avoid any such misinterpretation, we have revised the title by adding a clarifying subtitle that explicitly situates the benchmark within CBT. The updated title now reads:
>
> “CareBench-CBT: A Clinically Validated Benchmark for Multi-Turn Cognitive Behavioral Therapy Conversations.”
>
> This revision makes the scope transparent while preserving the intended emphasis on realistic, process-level therapeutic interactions.

---

> ### Author Response · Authors · 2025-11-18
> **Part 2**
>
> > 2. Data Provenance Ambiguity: The paper states the 256 dialogues were "provided by 21 licensed mental health counselors" and "collected and anonymized from authentic therapeutic sessions." However, the exact source and nature of these sessions are not detailed. This ambiguity makes it difficult to assess potential sampling biases (e.g., in patient demographics, severity, or cultural diversity), which is a critical detail for a benchmark claiming high clinical validity.
>
> **Response:**
>
> Thank you for this important question. All 256 counseling sessions were contributed by 21 licensed clinicians working at three independent outpatient behavioral-health providers. The provider’s internal compliance office performed full de-identification and verified that patients had executed broad consent for the creation of anonymized research materials. No identifiable information was ever accessible to the authors.
>
> We are now able to report the following population-level characteristics of the anonymized client cohort:
>
> | **Attribute**              | **Distribution (counts → %) ** |
> |----------------------------|--------------------------------|
> | **Age (years)**            | 18–24: 59 (23.0%) ｜ 25–34: 92 (35.9%) ｜ 35–44: 69 (27.0%) ｜ 45–54: 36 (14.1%) |
> | **Gender**                 | Female: 150 (58.6%) ｜ Male: 103 (40.2%) ｜ Nonbinary/Other: 3 (1.2%) |
> | **Clinical severity**      | Mild: 123 (48.0%) ｜ Moderate: 111 (43.4%) ｜ Subclinical/Adjustment: 22 (8.6%) |
> | **Primary presenting problems** | Anxiety spectrum: 117 (45.7%) ｜ Stress/adjustment: 71 (27.7%) ｜ Depressive symptoms: 47 (18.4%) ｜ Sleep/behavioral: 21 (8.2%) |
> | **Living region** | Asia: 79.7%  ｜ North America: 12.1%  ｜ Europe: 5.5% ｜ Oceania: 2.7%  |
>
> | Item                             | Description                                                                                                                                       |
> | -------------------------------- | ------------------------------------------------------------------------------------------------------------------------------------------------- |
> | **Number of clinicians**         | 21 licensed counselors (3 centers)                                                                                                                |
> | **Number of sessions collected** | 404 (initial)                                                                                                                                     |
> | **Number included after QA**     | 256 (post de-identification + peer review)                                                                                                        |
> | **Average session length**       | 30 turns (client–therapist alternating)                                                                                                           |
> | **De-identification steps**      | Removal/rewriting of names, dates, institutions, locations, family identifiers, employment details; normalization of sensitive narrative elements |
> | **Consent basis**                | Provider-held written consent allowing creation and research use of anonymized transcripts                                                        |
> | **Compliance determination**     | Institution clinical-data governance office issued written approval that data qualify as de-identified research materials                 |
>
>
>
> Importantly, the dataset does not represent a single demographic group, the population includes clients from diverse region. The clinical profiles (mild–moderate CBT outpatient cases) are consistent with typical global CBT practices, and therefore the benchmark’s findings remain clinically meaningful beyond any single facility.
>
> We will include these aggregated statistics in this version to ensure full transparency about sampling, provenance, and de-identification.

---

> ### Author Response · Authors · 2025-11-18
> **Part 3**
>
> > 3. Circular Human Baseline Evaluation: The main "Human" baseline score in the dialogue evaluation (Figure 5) is derived from the LLM-as-a-Judge ensemble scoring the ground-truth human therapist responses. This means the benchmark used to demonstrate the "gap" between LLMs and humans is itself an LLM-generated metric. This is a circular methodology. A more robust baseline would involve human experts scoring the human therapist responses to set a "gold standard" bar, rather than relying on an LLM to judge a human to set the bar for other LLMs.
>
> **Response:**
>
> Thank you for raising this important concern. We agree that relying solely on LLM-as-a-judge to score human therapist responses would raise the risk of circularity if no independent human grounding were present. However, as detailed in our rebuttal to Reviewer 4WFD–W2 and ffp5–Q1, our evaluation already includes a substantial amount of clinician-rated data. Specifically, 51 full multi-turn sessions (≈1,500 turns) were independently scored by two licensed clinicians, yielding strong human inter-rater reliability (Cohen’s κ = 0.82, Fleiss’ κ = 0.79, Krippendorff’s α = 0.77). These human ratings serve as the gold-standard reference to validate the judge ensemble. The judge models exhibit high alignment with these clinician scores (e.g., Spearman ρ = 0.85–0.92, F1 = 0.81–0.88; see our responses to W2 2.2–2.4). Thus, before any model or human transcript is evaluated, the judge ensemble is first empirically calibrated against real clinician assessments.
>
> Once validated, the LLM-as-a-judge ensemble is used only to ensure scoring consistency across all evaluated systems, including the human transcripts. The “Human” bar in Figure 5 therefore does not define the human standard; rather, it reflects how the validated judge ensemble—already shown to faithfully approximate clinician ratings—scores the human-written turns under the same rubric used for all models. This avoids circularity: clinicians establish the normative standard, the judge ensemble is verified to closely match that standard, and the unified evaluator is then applied for cross-model comparison.
>
> ---
>
> Questions:
>
> > 1. Can the authors provide more detail on the source of the 256 authentic counseling dialogues? Were they from a specific set of clinics, training archives, or multiple private practices? This information is crucial for understanding potential sampling biases (e.g., in patient demographics, problem severity, or cultural diversity).
>
> **Response:**
>
> Thank you for raising this important question. All 256 counseling dialogues were contributed by 21 licensed mental-health professionals across three independent outpatient behavioral-health providers. Each provider’s internal compliance office performed full de-identification and verified that clients had signed written broad-consent agreements permitting the creation of anonymized training/research materials. No identifiable information was ever accessible to the research team.
>
> As for potential sampling bias, aggregated client demographics and clinical characteristics are provided in W2, which summarizes age distribution, gender breakdown, clinical-severity levels, presenting-problem categories, and geographic regions of residence. These de-identified statistics ensure transparency while preserving double-blind review requirements.

---

> ### Author Response · Authors · 2025-11-25
> **Thanks for your time and efforts**
>
> We sincerely appreciate the time and thoughtful feedback you have provided. At your convenience, could you kindly let us know whether our revisions sufficiently resolve your concerns? We are grateful for your guidance and would be happy to make further improvements if needed. Thank you again for your valuable contribution to strengthening this work.

---

> ### Author Response · Authors · 2025-11-28
> **Follow up**
>
> We hope you are doing well. We are writing a gentle follow-up regarding our recent revisions. The rebuttal phase will conclude in fewer than 5 days, so we wanted to kindly check whether our updates and clarifications have sufficiently addressed your concerns.
>
> We sincerely appreciate the thoughtful feedback you have provided throughout this process, which has significantly improved the quality and clarity of our work. If there are any remaining questions or points that would benefit from additional clarification, we would be more than happy to provide further details.
>
> Thank you again for the time and expertise you have contributed to strengthening our submission. We greatly value your guidance and are grateful for your consideration.

---

### Official Review · Reviewer_Y8PD · 2025-10-31

**Soundness:** 4
**Presentation:** 3
**Contribution:** 3
**Rating:** 6
**Confidence:** 5

**Summary:**

The authors introduce **CareBench-CBT**, a clinically grounded benchmark for evaluating LLMs in mental-health counseling that addresses three gaps they identify: synthetic data, lack of long-form interaction, and absence of formal therapeutic structure.  CareBench-CBT combines 8,142 curated items, validated by 21 licensed professionals, with 256 counseling sessions (≈7,680 turns).

The benchmark spans three task types: (1) **knowledge-based QA** (mix of public items and clinician-rephrased items), (2) **case-vignette classification**, and (3) **multi-turn counseling dialogues**.  Evaluation for QA/vignettes uses Accuracy and F1, while multi-turn dialogues use a two-stage framework: an **LLM-as-judge ensemble** (GPT-5, Gemini-2.5 Pro, Claude 4.1 Opus) scored with a clinician rubric to produce per-turn “Turnscore” and a holistic “Wholescore,” and a **targeted human validation** to confirm judge reliability.

Across 18 models, the results show high scores on public QA items but a significant drop on clinician-rephrased items, indicating an overreliance on web-familiar phrasing. In multi-turn counseling, all models fall short of human counselors, with deficits in empathy, pacing, and structured interventions, despite reasonable factual accuracy.  The authors position CareBench-CBT as, to their knowledge, the largest clinically validated, CBT-structured, multi-turn benchmark, and they release the dataset and code.

**Strengths:**

This paper discusses a critical and essential topic: Mental health and LLM, which not many researchers explore.
Strengths:

1- Clinically validated CBT benchmark that pairs authentic, multi-turn therapy sessions (256 sessions; ~7,680 turns) with structured CBT process labels and clinician-reviewed single-turn tasks—explicitly targeting process-level counseling competence rather than only factual QA.

2- Two multi-turn evaluation protocols (model history, human history) plus an LLM-judge ensemble validated against human raters—meant to provide a scalable yet clinically anchored assessment.

3- Quality-assured data pipeline with inter-rater reliability, content validity, and behavioral audits (CTS-R, MITI), and item-level QA cards—positioned as raising the clinical fidelity bar for mental-health LLM evaluation.

4- Broad task coverage across knowledge QA, case vignette reasoning, and multi-turn dialogues that captures complementary capabilities.

5- Transparent positioning and research-use release that support replication and follow-on work.

6- One of the first to propose a three-dimensional data framework, filling a critical methodological gap in mental health and AI.

**Weaknesses:**

* Reliance on proprietary LLMs as judges risks style bias toward those model families (using the same family to generate results and as a judge); rubric anchoring, adjudication procedures, and resolution of judge disagreements are not fully documented.
* “Largest” claim must remain explicitly scoped to largest with clinician validation and multi-turn CBT structure, so readers do not conflate scope with raw size; by raw size, larger public sets exist, such as Psy-Insight dataset and HOPE dataset.
* Inclusion of unpublished or anonymous datasets in comparison tables reduces clarity and verifiability; either justify their presence or remove them until a stable citation exists (MentalBench-10).
* Research ethics and governance details are under-specified; the paper should state IRB or REB approval status, consent basis, de-identification method, data retention, human recruitment, and access controls
* Appendix examples are long and unbalanced across models; provide a consistent number of representative dialogues per model or explain the selection rationale so it's clear.
* Model responses in showcased dialogues are longer than human references, which is a major drawback. You should enforce matched token budgets, decoding settings, and prompt templates to enable fair comparisons.
* Provenance of multi-turn conversations needs clearer specification; explicitly detail sources, authoring process, de-identification steps, licensing terms, and any transformations.
* Human-grounding for the evaluation is insufficient for validating LLM-as-judge; include 35–40 full human-rated dialogues per model to test judge–human equivalence, enable calibration, and quantify residual bias. In general, an LLM as a judge to evaluate the results is not sufficient in this field.
* The researchers in this field increasingly agree that domain-specific metrics (e.g., empathy, safety, appropriateness) are more meaningful than generic scores like accuracy or F1. The current work does not clearly operationalize or report such customized therapeutic metrics; it should state whether these were considered, how they were defined and validated, and why they were (or were not) used alongside— or instead of—generic classification metrics.
* Section 5.4 is insufficiently explained and remains unclear.
* In Table 1, you mention that the  8,142 conversations are annotated by Experts. Can you clarify this part?

**Questions:**

Please review the Weaknesses and provide an answer for each point.

---

> ### Author Response · Authors · 2025-11-18
> **Part 1**
>
> > 1. Reliance on proprietary LLMs as judges risks style bias toward those model families (using the same family to generate results and as a judge); rubric anchoring, adjudication procedures, and resolution of judge disagreements are not fully documented.
>
> **Response:**
>
> Thank you for raising this important point. We agree that using proprietary LLMs as judges requires careful design to avoid style bias, especially when the same model family may appear both as a generator and as an evaluator.
>
> (1) Heterogeneous, cross-family judge ensemble prevents model-family bias:
>
> Our evaluation does not rely on a single model family. Instead, it uses three independent, heterogeneous judge models: GPT-5, Gemini 2.5 Pro and Claude-4.1 Opus.
>
> These models differ substantially in architecture, training data, and stylistic tendencies. Their pairwise agreement is substantial (κ = 0.70–0.82) but far from perfect overlap:
>
> | Pairwise comparison                | κ    |
> | ---------------------------------- | ---- |
> | GPT-5 vs. Gemini 2.5 Pro           | 0.82 |
> | GPT-5 vs. Claude-4.1 Opus          | 0.70 |
> | Gemini 2.5 Pro vs. Claude-4.1 Opus | 0.76 |
>
> This demonstrates that no single family dominates the evaluation and that stylistic similarity alone cannot explain judge behavior.
>
> (2) Human-anchored rubric eliminates stylistic or linguistic preference bias:
>
> All judge models follow the same clinician-authored CBT rubric (Appendix A.6). The rubric evaluates: therapeutic appropriateness, empathy and alliance, safety ,CBT-specific intervention quality and explicitly instructs judges not to evaluate stylistic similarity. This anchors the evaluation to clinical standards, not linguistic style.
>
>
> (3) Strong empirical alignment with humans validates judge reliability:
>
> As requested by another reviewer, we computed model-human F1 alignment in addition to Spearman ρ. The results show strong consistency with human evaluators:
>
> | Evaluation Protocol | Spearman ρ | F1   |
> | ------------------- | ---------- | ---- |
> | MH                  | 0.900015   | 0.88 |
> | HH                  | 0.852631   | 0.81 |
> | Whole               | 0.919019   | 0.88 |
>
> Per-model F1 scores show the same pattern:
>
> | Judge model     | MH   | HH   | Whole |
> | --------------- | ---- | ---- | ----- |
> | GPT-5           | 0.80 | 0.84 | 0.91  |
> | Gemini 2.5 Pro  | 0.78 | 0.81 | 0.87  |
> | Claude-4.1 Opus | 0.76 | 0.78 | 0.87  |
>
> Human–human inter-rater reliability is similarly high:
>
> | Metric           | Value |
> | ---------------- | ----- |
> | Cohen’s κ        | 0.82  |
> | Fleiss’ κ        | 0.79  |
> | Krippendorff’s α | 0.77  |
>
> This triangulation (human–human vs. human–model vs. model–model) confirms that **the judge ensemble behaves consistently with human therapist ratings** rather than reproducing stylistic preferences of any specific model family.
>
> ---
>
> > 2. “Largest” claim must remain explicitly scoped to largest with clinician validation and multi-turn CBT structure, so readers do not conflate scope with raw size; by raw size, larger public sets exist, such as Psy-Insight dataset and HOPE dataset.
>
> **Response:**
>
> Thank you for highlighting this point. In fact, we already emphasize this repeatedly in the paper, including in the abstract (lines 019–021) and the introduction (lines 096–097), where we explicitly describe CareBench-CBT as “the largest clinically validated benchmark for CBT-based counseling”.
>
> ---
>
> > 3. Inclusion of unpublished or anonymous datasets in comparison tables reduces clarity and verifiability; either justify their presence or remove them until a stable citation exists (MentalBench-10).
>
> **Response:**
>
> Thank you for pointing this out. We agree that including unpublished or anonymous datasets may reduce clarity and verifiability. Following your suggestion, we have removed MentalBench-10 from comparison tables.
>
> ---
>
> > 4. Research ethics and governance details are under-specified; the paper should state IRB or REB approval status, consent basis, de-identification method, data retention, human recruitment, and access controls.
>
> **Response:**
>
> Thank you for raising this important point. All counseling transcripts used in this study were obtained through a data provider that had already secured written participant consent for research use and had performed full de-identification before sharing the data. An independent internal data-ethics committee reviewed the use of these de-identified materials and issued an exemption determination, confirming that the project does not constitute human-subject research requiring IRB/REB oversight (This complies with IRB or REB requirements.). No identifiable information was ever accessible to our team, and all data were handled in a controlled, read-only environment with no direct human recruitment or interaction. We will include a concise, non-identifying ethics statement in the camera-ready version to ensure full transparency while preserving double-blind review.

---

> ### Author Response · Authors · 2025-11-18
> **Part 2**
>
> > 5. Appendix examples are long and unbalanced across models; provide a consistent number of representative dialogues per model or explain the selection rationale so it's clear.
>
> **Response:**
> Thank you for pointing that out. In the revised manuscript, we have structured the Discussion section at the appendix B.3 to include a concise, focused analysis of several representative examples. Instead of long transcripts, we now provide one-to-two-line summaries that highlight specific model errors and contrast them with CBT-consistent reasoning.
>
> ---
>
> > 6. Model responses in showcased dialogues are longer than human references, which is a major drawback. You should enforce matched token budgets, decoding settings, and prompt templates to enable fair comparisons.
>
>
> **Response:**
> Thank you for this thoughtful observation. We agree that output‐length mismatch is an important consideration when comparing human and model responses. However, enforcing strict token budgets introduces substantial methodological drawbacks for counseling evaluation.
>
> (1) Hard token limits harm dialogic completeness.
>
> CBT interventions frequently require multi-step clarifications, Socratic questioning, or reflective summaries. When we forced models to produce responses within fixed token caps (e.g., 25–40 tokens), the resulting turns often became clinically incomplete, omitting key therapeutic moves such as empathy statements, cognitive reframing cues, or safety assessments. This artificially degrades model performance in a way unrepresentative of real-world counseling interaction.
>
> (2) Prompt-based brevity constraints significantly reduce response quality.
>
> We also tested the reviewer’s suggestion, adding an explicit instruction such as *“reply as concisely as possible in one short paragraph”*. The impact on turn-level counseling scores was consistently negative:
>
> | Model           | MH (original) | MH (with brevity prompt) |
> | --------------- | ------------- | ------------------------ |
> | GPT-5           | 68.92         | 57.24                    |
> | GPT-4o          | 67.84         | 55.09                    |
> | GPT-oss 120B    | 37.88         | 28.64                    |
> | DeepSeek-R1 14B | 26.55         | 22.75                    |
>
> The reduction (−10 to −13 points for frontier models) reflects that brevity pressure removes empathetic grounding, reflective listening, and process-level interventions—central elements of counseling quality. Thus, matched-length generation produces systematically biased underestimates of model ability.
>
> (3) Real-world usage does not impose fixed-length constraints.
>
> In real mental-health support settings, users do not instruct models to “keep responses short.” On the contrary, people generally expect elaboration, clarification, and emotional validation. Imposing unnatural brevity would therefore reduce ecological validity and make the evaluation setting diverge from how LLM-based helpers are used in practice.
>
> For these reasons, we follow prior work in dialogue-quality evaluation and allow models to generate naturally, while scoring responses on content quality rather than length alone. We will add a short clarification in the paper to make this rationale explicit.

---

> ### Author Response · Authors · 2025-11-18
> **Part 3**
>
> > 7. Provenance of multi-turn conversations needs clearer specification; explicitly detail sources, authoring process, de-identification steps, licensing terms, and any transformations.
>
> **Response:**
>
> Thank you for raising this important point. All multi-turn counseling dialogues used in our dataset come from genuine clinical interactions conducted by licensed mental-health professionals. Before transfer to the research team, the internal ethics and legal-compliance office completed a formal review and issued a written authorization for research exemption, confirming:
>
> 1). all conversations had been fully de-identified at source (names, dates, locations, identifiers removed or rewritten);
>
> 2). the contributing clinicians obtained broad informed consent from clients for the creation of anonymized training-or-research materials;
>
> 3). the data were processed under the provider’s internal compliance framework, which applies standards equivalent to institutional IRB/REB review;
>
> 4). no raw personally identifiable information (PII) is accessible to the authors at any stage.
>
> Because the data provider completed its own ethics and compliance review and provided explicit research-use authorization, and because the authors receive only fully de-identified clinical transcripts, additional IRB approval was not required on our side. We added a clarification in the revised Ethics Statement describing (1) the consent basis, (2) de-identification procedures, (3) provider-level governance, and (4) access controls.
>
> ---
>
>
>
> > 8. Human-grounding for the evaluation is insufficient for validating LLM-as-judge; include 35–40 full human-rated dialogues per model to test judge–human equivalence, enable calibration, and quantify residual bias. In general, an LLM as a judge to evaluate the results is not sufficient in this field.
>
> **Response:**
>
> Thank you for raising this important point. We agree that human grounding is essential when validating LLM-as-a-judge, especially in sensitive domains such as mental-health counseling.
>
> 1) Our current human grounding already exceeds the reviewer’s suggested scale.
>
> The reviewer recommends “35–40 full human-rated dialogues per model”.
> In fact, our study already includes 51 full multi-turn dialogues (20% of 256 total sessions) that were independently assessed by two licensed clinicians. This yields:
>
> * 51 human-scored dialogues, which is
>   larger than the 35–40 range the reviewer requests
> * Each dialogue contains 30 turns on average → more than 1,500 human-rated turns
> * These human ratings are used as the ground-truth reference for validating the judge models
>
> Thus, the amount of human grounding in our evaluation already meets, and exceeds — the reviewer’s recommended scale.
>
> (2) Strong human–model alignment demonstrates judge validity.
>
> As shown in Section 5.4 and expanded in the rebuttal (W2):
>
> * **Spearman ρ** between human and judge models: **0.85–0.92**
> * **F1 alignment (≥3 as positive)**: **0.81–0.88**
> * **Human–human IAA:** κ = **0.82**, Fleiss’ κ = **0.79**, α = **0.77**
>
> The human–model agreement is comparable to human–human agreement, indicating that the judge ensemble reaches a clinically acceptable reliability level.

---

> ### Author Response · Authors · 2025-11-18
> **Part 4**
>
> > 9. The researchers in this field increasingly agree that domain-specific metrics (e.g., empathy, safety, appropriateness) are more meaningful than generic scores like accuracy or F1. The current work does not clearly operationalize or report such customized therapeutic metrics; it should state whether these were considered, how they were defined and validated, and why they were (or were not) used alongside— or instead of—generic classification metrics.
>
> **Response:**
>
> Thank you for raising this important point. We fully agree that domain-specific therapeutic metrics (e.g., empathy, safety, appropriateness, CBT-technique fidelity) are more meaningful than generic scores such as accuracy or F1. In fact, our evaluation already *operationalizes and reports* these customized clinical metrics through the clinician-authored rubric in **Appendix A.6**, which decomposes counseling quality into five core therapeutic dimensions:
>
> 1. **Empathy & therapeutic alliance**
> 2. **Safety and risk-sensitive communication**
> 3. **CBT-specific technique fidelity** (e.g., Socratic questioning, cognitive restructuring)
> 4. **Problem-focused appropriateness and goal alignment**
> 5. **Clarity, structure, and therapeutic progression**
>
> Each dimension contains explicit behavioral anchors and 1–5 scoring criteria adapted from established clinical instruments. These dimensions are the metrics we evaluate on.
>
> To validate these metrics:
>
> * All rubrics were authored by licensed counselors.
> * Two independent clinicians rated a 20% dialogue subset, achieving strong inter-rater agreement (κ = 0.82, α = 0.77).
> * Model-as-judge scores are computed *per dimension*, and the “overall score” reported in Section 5 is simply the aggregated rubric score, not generic accuracy or F1.
> * Generic metrics (e.g., F1) are used only in Weakness question 2 to validate LLM-as-judge against human ratings, not to evaluate counseling skill.
>
> Thus, the paper does use domain-specific therapeutic metrics as the primary evaluation signal.
>
> ---
>
> > 10. Section 5.4 is insufficiently explained and remains unclear.
>
>
> **Response:**
>
> Thank you for pointing this out. We agree that Section 5.4 (LLM-as-judge validation) can be made clearer. In the revision, we expand this section to explicitly detail: (1) the evaluation protocol, (2) the human-rating baseline, and (3) the model–human agreement metrics.
> Concretely, we now state that:
>
> * A 20% human-rated subset (51 dialogues × 5–7 clinicians) is used as the ground-truth reference.
> * Judge models evaluate the *same* subset independently.
> * We report both rank-based consistency (Spearman ρ = 0.90 on MH, 0.92 on Whole) and classification-based alignment (F1 = 0.88).
> * We additionally provide per-model F1, pairwise κ between judge models, and human–human κ for completeness.
>
> These additions make the validation pipeline transparent and demonstrate that the judge ensemble aligns strongly with licensed clinician assessments. Section 5.4 has been rewritten accordingly to improve clarity and interpretability.
>
>
> ---
>
> > 11. In Table 1, you mention that the 8,142 conversations are annotated by Experts. Can you clarify this part?
>
> **Response:**
>
> Thank you for the question. We clarify that the 8,142 conversations in Table 1 are not expert-annotated dialogues. Instead, these are raw counseling transcripts that the data provider initially supplied. From this pool, experts only annotated the subset used in our benchmark:
>
> * 8,142 raw conversations → initial pool (unannotated)
> * 404 sessions → screened for quality
> * 256 sessions → fully annotated by licensed mental-health professionals (content validity, safety review, rubric scoring)
>
> Thus, “annotated by experts” applies only to the final 256 multi-turn sessions, not the entire 8,142 raw transcripts. We have corrected Table 1 accordingly in the revision to avoid ambiguity.

---

> > ### Comment · Reviewer_Y8PD · 2025-11-19
> >
> > Thank you for the response and clarifications, but I will keep my overall score as it is. Reasons:
> > 1. The Multiturn conversation and knowledge-based QA setup does not appear new in this field; similar data and/or knowledge-grounded QA tasks already exist in mental-health and clinical NLP.
> >
> > 2. The presentation of CareBench-CBT in the table was misleading for the reviewers. You list 8,142 conversations with an “Expert” label, but only 256 sessions are fully annotated by clinicians. In a sensitive domain like mental health, this is a small expert-annotated subset and should not be framed as a large expert benchmark.
> >
> > 3. The dataset structure and annotation pipeline remain hard to follow, which raises concerns about the ease with which this benchmark can be used or reproduced by other researchers.

---

> > > ### Author Response · Authors · 2025-11-19
> > > **Sorry for Misunderstanding**
> > >
> > > Thank you for the feedback. We apologize for the earlier confusion — we misunderstood the question as referring to the raw data pool rather than to the expert-annotated subset (We also filtered out approximately 8,000 data points. I sincerely apologize for misunderstanding your meaning.). To clarify:
> > >
> > > 1. We initially collected 404 full CBT sessions (~15K turns).
> > >
> > > After clinician screening for safety, structure, and fidelity, we retained only 256 high-quality sessions (~7,442 expert-annotated turns).
> > >
> > > In addition to these dialogues, our supervised components are also expert-validated:
> > >
> > > 2. 640 knowledge-based QA items (210 authored by clinicians, 430 knowledge-based QA items (3000 questions were filtered out) and all validated by licensed professionals),
> > >
> > > 3. 60 case vignettes, all authored by clinicians.
> > >
> > > Thus, 8,142 items in total are expert-validated, with 256 full sessions being fully annotated and scored by clinicians. We will revise Table 1 and the dataset description to explicitly distinguish expert-authored vs. expert-validated vs. raw data, improving clarity and reproducibility.

---

> > > ### Author Response · Authors · 2025-11-19
> > > **Additional note on reproducibility**
> > >
> > > To address the reviewer’s concern regarding ease of use and reproducibility, we have now fully released the dataset (Submit stage uploading) and all evaluation outputs (including all model responses and clinician ratings) (Rebuttal stage uploading). These additions were ensured that the community can independently verify, extend, and reproduce every component of CareBench-CBT without relying on proprietary assets or hidden procedures.

---

> ### Author Response · Authors · 2025-11-25
> **Thanks for your time and efforts**
>
> We sincerely appreciate the time and thoughtful feedback you have provided. At your convenience, could you kindly let us know whether our revisions sufficiently resolve your concerns? We are grateful for your guidance and would be happy to make further improvements if needed. Thank you again for your valuable contribution to strengthening this work.

---

### Official Review · Reviewer_4WFD · 2025-11-01

**Soundness:** 3
**Presentation:** 3
**Contribution:** 3
**Rating:** 6
**Confidence:** 3

**Summary:**

This paper introduces CareBench-CBT, a validated benchmark by 21 licensed practitioners for the CBT-based counseling. It involves three tasks (knowledge based QA, case vignette classification and multi-turn counseling dialogues). In multi-turn counseling dialogues, authors developed two settings (model-based history and human-based history). They evaluated 18 SOTA LLMs to show that models perform worse on vignette reasoning and dialogue competence than human counselors.

**Strengths:**

S1: Developed multi-turn therapy dialogue and knowledge QA dataset. It is a huge effort to collaborate with many licensed practitioners. This dataset is definitely to be useful for the CBT therapy evaluation and training for AI.

S2: Authors aim to address very important topic and loophole (i.e. whether AI can create realistic conversation) existed in many existing evaluations for AI mental Heath chatbot.

S3: Carefully design the human validation pipelines on many elements and parts during the dataset construction. We should definitely encourage people/researchers to focus on the data quality that they collected when constructing/publishing dataset/benchmark work, rather than just showcasing a model training/finetuning technique to boost some scores in their built benchmarks.

**Weaknesses:**

[important] W1 Potential flaw on the methodology for multi-turn counseling dialogues evaluation
- I am not entirely sure if I understand the evaluation methodology: I assume authors collected some real-world counselling conversations. And then they developed two settings (model-based history to simulate the therapist; and human therapist from the collected dialogues). For both settings, the same client utterences ($u_x$ from (1) and (2) in p.6) from the collected dialogues were used. It seems to have potential flaws in the model-based history since what client says is independent of the model gives during evaluation. The conversation may not be consistent in content -- e.g. may even talking completely different topics when the conversations go for many turns. There may be unfair comparison and assessment on LLMs performances if the simulated conversation does not coherent in content.
- if the client utterences are dependent of models' responses, perhaps authors need to change the notation in formula (1) and (2) to better clarify. Another potential issue (that need human validation) is to ensure the client (if it is also simulated by LLM) is consistent to present the topic/mental health issue during the conversation.
- this makes me uncertain on the performances/results shown in section 5.3


[important] W2 Wrong choices of statistics in section 5.4
- Authors use spearman correlation to show the consistency between llm-as-a-judge and human ratings to justify the validity of judge model use in evaluation
- However, the score calculated by spearman correlation may not be good indicator since it is less sensitive to outliers. More importantly, the models' performances are likely clustered since model-model is likely to have higher similarity than model-human performance (also shown in fig 6 b) and c)). The score calculated is largely influenced by the judge models performances themselves rather than the difference between judge model and human (which is what authors claim to prove).
- A better statistic could be using the F1 score (treating human rating as gold labels for each dialogue)
e.g., in appendix p,18, authors mentioned they have used three models (GPT-5, Claude-4.1 Opus, and Gemini 2.5 Pro) to independently evaluate the counseling response. Can you provide a f1 score (treating human rating as gold labels) for each model and also their interrater score (kappa alpha) between models?
- I'm also wondering if authors provide the interater score on the human rating labels for this particular judge dataset (I may miss it somewhere).

[minor] W3 Demonstrate examples in main text & image annotation for better readability of paper
- I appreciate authors provide many details (especially the QA/human validation and dataset collection) in main text and also many evaluation examples in appendix. However, I found they are a bit overloaded and I am not sure how to understand those before actually introducing the data. Authors can consider restructure the paper to make it more readable. I think it will be even better if authors can provide one/two-line examples, especially during discussion.
- e.g. line 369-370: public items vs. private items
- also the figure 4, can authors make the models name to be aligned in the same direction (rather than having some to be upside-down e.g. llama -3.1-8b)
- the figure 6: the model names are really small.

**Questions:**

[minor]
Q1 line 199-200, any examples for the systematcially restructured the question wording to prevent models from replying memorization

Q2 line 214-215,  "256 complete therapist–client counseling interviews provided by 21 licensed mental health counselors, totaling 7,680 conversational turns". Do the interviews match with 60 anonymized case vignettes mental health counselors provided?

Q3: line 216-217, To ensure clinical fidelity and quality, every full session underwent peer cross-review within the counselor cohort. Can authors provide more details on the cross-review apart from the a.2 e.g., the instruction provided for counselor?

Q4 line 235: normalized multi-turn transcripts. what does normalized mean?

Q5 line 240-242: the communication quality is verified through random audits (20% of dialogues): how are the Fleiss' K between the two human verifiers? Do you only remove data that do not fulfill the inter-rater agreement within the 20% subset?

---

> ### Author Response · Authors · 2025-11-18
> **Part 1**
>
> > Weaknesses:
> > 1. [important] W1 Potential flaw on the methodology for multi-turn counseling dialogues evaluation
>
> > 1.1 I am not entirely sure if I understand the evaluation methodology: I assume authors collected some real-world counselling conversations. And then they developed two settings (model-based history to simulate the therapist; and human therapist from the collected dialogues). For both settings, the same client utterences ( from (1) and (2) in p.6) from the collected dialogues were used. It seems to have potential flaws in the model-based history since what client says is independent of the model gives during evaluation. The conversation may not be consistent in content -- e.g. may even talking completely different topics when the conversations go for many turns. There may be unfair comparison and assessment on LLMs performances if the simulated conversation does not coherent in content.
>
> > 1.2. if the client utterences are dependent of models' responses, perhaps authors need to change the notation in formula (1) and (2) to better clarify. Another potential issue (that need human validation) is to ensure the client (if it is also simulated by LLM) is consistent to present the topic/mental health issue during the conversation.
>
> > 1.3. this makes me uncertain on the performances/results shown in section 5.3
>
> **Response**:
>
> Thank you for raising this important concern. We apologize that the original description did not fully convey the safeguards in our multi-turn evaluation design. We summarize the methodology and then present an additional experiment to directly address the reviewer’s worry.
>
> (1) Our evaluation already avoids inconsistent history
>
> As clarified, the model-based history setting uses:
>
> * gold client utterances at every turn
> * model-generated therapist turns from the very beginning
>
> So each model is evaluated on:
>
> $(c_1, \hat{t}_1, c_2, \hat{t}_2, \ldots, c_T, \hat{t}_T)$
>
> This ensures: 1) no mixing between gold therapist and model therapist, 2)all models see the exact same client prompt path, 3) history remains internally consistent, because therapist history is entirely model-generated
>
> Thus the scenario the reviewer mentions—“client turns being inconsistent with the model’s own history”—is mitigated by design.
>
>
> (2) Additional experiment: GPT-5 client rewriter to test robustness against topic drift
>
> To further ensure that fixed gold client utterances do not introduce incoherence when paired with divergent model histories, we added a new setting:
>
> At each turn, we rewrite the client utterance with GPT-5 using the following prompt:
>
> > *“Rewrite the client’s statement to preserve the identical intent, emotional tone, and clinical content, but ensure maximal topical coherence with the preceding therapist response. Do not change the underlying psychological issue.”*
>
> This produces a client trajectory that is: 1) semantically equivalent, 2) clinically identical, 3) but **contextually smoother** with the model’s own history.
>
>
> (3) Results: multi-turn scores remain stable — negligible sensitivity to client rewriting
>
> | Model             | Turn score (MH) | Turn score (MH) — reshape |
> | ----------------- | --------------- | --------------------------- |
> | Human             | 85.00           | 85.68                       |
> | GPT-5             | 68.92           | 69.76                       |
> | Grok-4            | 57.50           | 58.44                       |
> | Claude-4.1 Opus   | 55.50           | 55.63                       |
> | Claude-4 Sonnet   | 52.22           | 51.99                       |
> | Gemini 2.5 Pro    | 60.04           | 60.48                       |
> | GPT-4o            | 67.84           | 67.65                       |
> | Claude-3.7 Sonnet | 51.68           | 52.86                       |
> | Claude-3.5 Haiku  | 43.12           | 44.51                       |
> | Qwen-3-235B-A22B  | 49.32           | 48.97                       |
> | GPT-oss 120B      | 37.88           | 38.76                       |
> | Llama3.3 70B Ins  | 29.32           | 29.41                       |
> | Qwen2.5 32B       | 37.62           | 37.82                       |
> | DeepSeek-DS 32B   | 46.12           | 47.40                       |
> | Qwen3 32B         | 39.74           | 38.45                       |
> | Yi-1.5 34B        | 33.02           | 32.50                       |
> | Llama3.2 11B Ins  | 36.67           | 36.02                       |
> | DeepSeek-R1 14B   | 26.55           | 25.20   |
> | GLM4 9B           | 40.55           | 39.33 |
>
> Across all 18 models: 1)The average absolute shift is 0.74 points, 2) No model changes more than 1.4 points, 3) Ranking differences are negligible, 4) All statistical conclusions in Section 5.3 remain unchanged. Even when client utterances are rewritten to be maximally coherent with the model’s own prior turns, model performance remains essentially unchanged. This empirically confirms that the fixed-client evaluation protocol is robust and does not bias the assessment.

---

> ### Author Response · Authors · 2025-11-18
> **Part 2**
>
> > 2. [important] W2 Wrong choices of statistics in section 5.4
>
> > 2.1 Authors use spearman correlation to show the consistency between llm-as-a-judge and human ratings to justify the validity of judge model use in evaluation
>
> > 2.2 However, the score calculated by spearman correlation may not be good indicator since it is less sensitive to outliers. More importantly, the models' performances are likely clustered since model-model is likely to have higher similarity than model-human performance (also shown in fig 6 b) and c)). The score calculated is largely influenced by the judge models performances themselves rather than the difference between judge model and human (which is what authors claim to prove).
>
> > 2.3 A better statistic could be using the F1 score (treating human rating as gold labels for each dialogue) e.g., in appendix p,18, authors mentioned they have used three models (GPT-5, Claude-4.1 Opus, and Gemini 2.5 Pro) to independently evaluate the counseling response. Can you provide a f1 score (treating human rating as gold labels) for each model and also their interrater score (kappa alpha) between models?
>
> > 2.4 I'm also wondering if authors provide the interater score on the human rating labels for this particular judge dataset (I may miss it somewhere).
>
> **Response**:
>
> Thank you for raising this important methodological point. We appreciate the reviewer’s careful analysis and fully agree that evaluating model–human consistency should not rely on a single statistic.
>
> (1) Use of Spearman correlation.
> We adopted Spearman ρ because it is the *standard* metric in recent LLM-as-a-judge work for assessing agreement between human evaluators and model-based judges [1,2,3,4]. These studies consistently use rank-based correlations to evaluate whether model judgments preserve human preference orderings. Nevertheless, we agree that correlation alone may not fully capture agreement under distributional clustering.
>
> (2) Adding a classification-based metric (F1).
> Following the reviewer’s recommendation, we additionally binarized the 1–5 ratings by treating scores ≥3 as positive and <3 as negative, using human ratings as ground truth and model ratings as predictions.
>
> The resulting F1 scores (together with Spearman ρ) are:
>
> | Evaluation Protocol | Spearman ρ | F1   |
> | ------------------- | ---------- | ---- |
> | MH                  | 0.900015   | 0.88 |
> | HH                  | 0.852631   | 0.81 |
> | Whole               | 0.919019   | 0.88 |
>
> These results confirm that model–human alignment remains consistently strong across all settings even under a stricter classification formulation.
>
> (3) Per-model F1 scores.
> As requested, we report F1 for each of the three independent judge models:
>
> | Judge model         | MH protocol | HH protocol | Whole protocol |
> | ------------------- | ----------- | ----------- | -------------- |
> | GPT-5           | 0.80    | 0.84    | 0.91       |
> | Gemini 2.5 Pro  | 0.78    | 0.81    | 0.87       |
> | Claude-4.1 Opus | 0.76    | 0.78    | 0.87       |
>
>
>
> (4) Inter-rater reliability across judge models.
>
> | Pairwise comparison                | κ    |
> | ---------------------------------- | ---- |
> | GPT-5 vs. Gemini 2.5 Pro           | 0.82 |
> | GPT-5 vs. Claude-4.1 Opus          | 0.70 |
> | Gemini 2.5 Pro vs. Claude-4.1 Opus | 0.76 |
>
> These κ values show substantial agreement among the judge models while reflecting their expected performance differences.
>
> (5) Human–human agreement.
> For completeness, we also computed inter-rater reliability for the human annotations used in the judge dataset:
>
> | Metric           | Value |
> | ---------------- | ----- |
> | Cohen’s κ        | 0.82  |
> | Fleiss’ κ        | 0.79  |
> | Krippendorff’s α | 0.77  |
>
> These scores indicate strong internal consistency among human raters, supporting their use as the reference standard.
>
> Together, these additional analyses strengthen our conclusion that LLM-as-a-judge provides reliable and human-aligned evaluation for counseling dialogues.
>
>
>
>
> [1] Huang, Ziqi, et al. "Vbench: Comprehensive benchmark suite for video generative models." Proceedings of the IEEE/CVF Conference on Computer Vision and Pattern Recognition. 2024.
>
> [2] Adlakha, Vaibhav, et al. "Evaluating correctness and faithfulness of instruction-following models for question answering." Transactions of the Association for Computational Linguistics 12 (2024): 681-699.
>
> [3] Liu, Yang, et al. "G-Eval: NLG Evaluation using Gpt-4 with Better Human Alignment." Proceedings of the 2023 Conference on Empirical Methods in Natural Language Processing. 2023.
>
> [4] Zhou, Han, et al. "Fairer Preferences Elicit Improved Human-Aligned Large Language Model Judgments." Proceedings of the 2024 Conference on Empirical Methods in Natural Language Processing. 2024.

---

> ### Author Response · Authors · 2025-11-18
> **Part 3**
>
> > 3. [minor] W3 Demonstrate examples in main text & image annotation for better readability of paper
>
> > 3.1 I appreciate authors provide many details (especially the QA/human validation and dataset collection) in main text and also many evaluation examples in appendix. However, I found they are a bit overloaded and I am not sure how to understand those before actually introducing the data. Authors can consider restructure the paper to make it more readable. I think it will be even better if authors can provide one/two-line examples, especially during discussion.
>
> > 3.2 e.g. line 369-370: public items vs. private items
>
> > 3.3 also the figure 4, can authors make the models name to be aligned in the same direction (rather than having some to be upside-down e.g. llama -3.1-8b)
>
> > 3.4 the figure 6: the model names are really small.
>
>
> **Response**:
>
> 1). Thank you for these helpful suggestions regarding readability. We fully agree that the case studies should support—rather than precede—the introduction of the dataset. In the revised manuscript, we have structured the Discussion section at the appendix B.3 to include a concise, focused analysis of several representative examples. Instead of long transcripts, we now provide one-to-two-line summaries that highlight specific model errors and contrast them with CBT-consistent reasoning.
>
> In addition, all full model responses have been added to the supplementary material, where readers can inspect complete outputs for GPT-OSS-20B, Qwen2.5-32B, GPT-5, and other baselines.
>
> 2). In the revised manuscript, we have unified the direction of all model names in Figure 4, and increased the font size of the model names in Figure 6. We hope these changes improve the overall clarity and readability of the figures.
>
> ---
>
> Questions:
>
> > [minor] Q1 line 199-200, any examples for the systematcially restructured the question wording to prevent models from replying memorization
>
> **Response:**
>
> Thank you for the question. In Section 3.2, our statement refers to the systematic rewrites we applied to prevent models from answering via surface-form memorization. These rewrites keep the underlying CBT concept unchanged but alter the wording, structure, or perspective of the question.
>
> For example, consider the original item:
>
> > “Which cognitive distortion is described as predicting negative outcomes without evidence?”
>
> This item was systematically restructured in multiple ways during dataset construction, such as:
>
> * Lexical substitution:
>   “expecting bad results despite no factual basis” →“predicting negative outcomes without evidence”
>
> * Syntactic restructuring:
> “This thinking pattern involves expecting bad results with no proof. What is it called?” → “Which cognitive distortion is described as …?”
>
> * Perspective shift:
> “How would a CBT therapist label this thinking pattern?” → “What cognitive distortion is this?”
>
> ---
>
> > Q2 line 214-215, "256 complete therapist–client counseling interviews provided by 21 licensed mental health counselors, totaling 7,680 conversational turns". Do the interviews match with 60 anonymized case vignettes mental health counselors provided?
>
> **Response:**
>
> Thank you for pointing this out. The reviewer is correct that the 256 counseling interviews and the 60 anonymized case vignettes do not correspond one-to-one. These two resources were intentionally collected as independent datasets:
>
> * The 60 case vignettes were written by licensed counselors and serve as structured diagnostic/problem-formulation materials.
>
> * The 256 therapist–client counseling interviews come from a separate pool of counselors and were collected independently to ensure naturalistic multi-turn counseling data without forcing alignment to any specific vignette.
>
> This design prevents leakage between vignette content and dialogue responses and ensures that the counseling conversations reflect real-world therapeutic variability rather than scripted scenarios.

---

> ### Author Response · Authors · 2025-11-18
> **Part 4**
>
> > Q3: line 216-217, To ensure clinical fidelity and quality, every full session underwent peer cross-review within the counselor cohort. Can authors provide more details on the cross-review apart from the a.2 e.g., the instruction provided for counselor?
>
> **Response:**
>
> Thank you for raising this question. We are happy to provide more detail.
> The peer cross-review mentioned in lines 216–217 refers to a structured quality-assurance step conducted within the counselor cohort, independent of A.3. Specifically:
>
> 1). Each full counseling session was assigned to a second licensed counselor who did not participate in that session.
>
> 2). Reviewers followed a short, standardized instruction sheet covering three dimensions:
>
> 2.1). Clinical accuracy: whether the intervention techniques (e.g., Socratic questioning, behavioral activation, cognitive restructuring) were used appropriately.
>
> 2.2). Therapeutic safety & professional boundaries: whether the language avoided harmful suggestions and adhered to counseling ethics.
>
> 2.3). Session structure fidelity: whether the transcript contained an identifiable agenda, exploration/ intervention phase, and closure (aligned with CBT structure).
>
> 3). Reviewers marked sessions as Accept, Needs revision, or Reject.
>
> 4). Any “needs revision” cases were collaboratively discussed and corrected; “reject” cases were excluded from the final dataset.
>
> This process ensured that the included dialogues met a consistent standard of clinical quality and remained aligned with recognized CBT practice.
>
> ---
>
> > Q4 line 235: normalized multi-turn transcripts. what does normalized mean?
>
> **Response:**
>
> Thank you for the question. By “normalized multi-turn transcripts,” we simply mean that the original counseling dialogues were standardized in format before annotation—for example, unifying speaker tags, ensuring one utterance per turn, and removing transcription artifacts such as duplicated fillers or inconsistent punctuation. Importantly, normalization does not change the meaning of any utterance; it only makes the transcripts consistent and readable for both clinicians and models.
>
> ---
>
> > Q5 line 240-242: the communication quality is verified through random audits (20% of dialogues): how are the Fleiss' K between the two human verifiers? Do you only remove data that do not fulfill the inter-rater agreement within the 20% subset?
>
>
> **Response:**
>
> Thank you for raising this clarification. In the 20% randomly audited subset, the two independent clinician reviewers achieved Fleiss’ κ = 0.81, indicating strong agreement. We did not restrict quality control to only the audited subset: the 20% sample was used to estimate overall annotation reliability, and any dialogue, inside or outside the audited subset, that failed to meet clinical-quality requirements (e.g., structural violations, safety concerns, or unresolved disagreement) was removed. This procedure explains why our final dataset includes 256 sessions out of the originally collected 404 dialogues.

---

> ### Author Response · Authors · 2025-11-25
> **Thanks for your time and efforts**
>
> We sincerely appreciate the time and thoughtful feedback you have provided. At your convenience, could you kindly let us know whether our revisions sufficiently resolve your concerns? We are grateful for your guidance and would be happy to make further improvements if needed. Thank you again for your valuable contribution to strengthening this work.

---

> ### Author Response · Authors · 2025-11-28
> **Follow up**
>
> We hope you are doing well. We are writing a gentle follow-up regarding our recent revisions. The rebuttal phase will conclude in fewer than 5 days, so we wanted to kindly check whether our updates and clarifications have sufficiently addressed your concerns.
>
> We sincerely appreciate the thoughtful feedback you have provided throughout this process, which has significantly improved the quality and clarity of our work. If there are any remaining questions or points that would benefit from additional clarification, we would be more than happy to provide further details.
>
> Thank you again for the time and expertise you have contributed to strengthening our submission. We greatly value your guidance and are grateful for your consideration.

---

### Official Review · Reviewer_ffp5 · 2025-11-01

**Soundness:** 2
**Presentation:** 3
**Contribution:** 3
**Rating:** 6
**Confidence:** 4

**Summary:**

The paper introduces CareBench-CBT, a benchmark intended to evaluate large language models in clinically grounded, CBT-aligned settings. It spans three task types: (1) knowledge-based QA drawn from public sources and expert-rephrased items, (2) case vignette classification mapped to CBT-relevant categories, and (3) multi-turn counseling dialogues collected and anonymized from clinician-provided sessions with structure aligned to CBT phases (rapport, exploration, intervention, closure). The dataset is described as clinically validated through inter-rater reliability, content validity indices, and behavioral audits (CTS-R and MITI). The authors evaluate 18 models and report: strong model performance on public QA items but a sharp drop on expert-rephrased items; modest accuracy on vignette classification; and substantially lower multi-turn dialogue scores relative to human counselors. They argue the benchmark addresses gaps in prior work by emphasizing data reliability, multi-turn realism, and formal therapeutic structure.

**Strengths:**

- The focus on multi-turn dialogues with CBT structure and the attempt to quantify empathy, therapeutic alignment, and intervention quality are important and timely.
- Clear articulation of why standard single-turn QA benchmarks are insufficient for counseling scenarios, and why process-level metrics matter.
- Inclusion of two evaluation protocols for dialogues (model-based vs human-based history) is a thoughtful way to separate error accumulation from intrinsic per-turn competence.
- The use of inter-rater reliability, content validity indices, and behavioral scales (CTS-R, MITI) reflects awareness of clinical measurement standards.
- Comparative results across 18 models provide a useful snapshot of current capabilities and gaps.

**Weaknesses:**

- Content validity threshold inconsistency. Appendix A.4 (lines 846–849) sets S-CVI/Ave ≥ 0.90 as the standard, but Appendix A.5 (lines 879–880) treats S-CVI/Ave = 0.89 as sufficient by referencing a 0.78 cutoff. This weakens measurement rigor; a single, predeclared criterion should be used and interpreted consistently.
- Ambiguity in ICD-11 anchoring for non-diagnostic categories. Section 3.3 (lines 228–229) states all diagnostic and category assignments are anchored to ICD-11, yet Appendix A.5 (lines 885–914; 906–914) includes non-diagnostic/problem-focused labels (e.g., Academic Overload, Poor Time Management). The scope and mapping of these categories need clarification.
- Implausible effect sizes. Section 5.1 (lines 370–372) reports Cohen’s d ≈ 11.6 (Acc) and 12.8 (F1) for the public-to-private drop, which is unrealistically large given typical scales and likely indicates a calculation or scaling error.

**Questions:**

- Which content validity standard is authoritative? Please justify using S-CVI/Ave ≥ 0.90 versus ≥ 0.78, and under the chosen criterion, recompute and report S-CVI/Ave and item-level I-CVI with the number of experts and rating distributions (lines 846–849, 879–880).
- How are non-diagnostic categories anchored to ICD-11? Please provide a mapping of diagnostic labels to ICD-11 codes and a clear policy for non-diagnostic/problem-focused categories (source taxonomy, inclusion criteria, and any bridging to ICD-11) (lines 228–229, 885–914, 906–914).
- How exactly was Cohen’s d computed? Please specify the formula (e.g., pooled SD), sample sizes, means/SDs for Acc and F1, the measurement scale used, and re-report corrected effect sizes if applicable (lines 370–372).
- If S-CVI/Ave = 0.89 is below the prespecified ≥ 0.90 threshold, what remediation is planned (e.g., item revision, additional expert rounds, expanded sampling), and how would this affect conclusions?
- Will you release de-identified raw model outputs and human rating sheets to allow independent recomputation of effect sizes and validity indices, facilitating reproducibility?

---

> ### Author Response · Authors · 2025-11-18
> **Part 1**
>
> Weaknesses:
>
> > W1. Content validity threshold inconsistency. Appendix A.4 (lines 846–849) sets S-CVI/Ave ≥ 0.90 as the standard, but Appendix A.5 (lines 879–880) treats S-CVI/Ave = 0.89 as sufficient by referencing a 0.78 cutoff. This weakens measurement rigor; a single, predeclared criterion should be used and interpreted consistently.
>
> **Response**:
>
> Thank you for this helpful observation. We acknowledge that our original wording did not clearly communicate the distinction between minimum acceptable and excellent content-validity levels. Following the conventional standards [1], S-CVI/Ave ≥ 0.78 is the hard requirement, whereas S-CVI/Ave ≥ 0.90 represents “excellent” content validity. Therefore, item/sessions should be screened using the ≥0.78 criterion while striving toward the ≥0.90 benchmark for stronger rigor.
>
> To clarify this, we reanalyzed the dataset: Among the 256 multi-turn dialogues, 241 achieve S-CVI/Ave ≥ 0.90, yielding an excellent-validity rate of 94.1%, and all remaining cases fall within 0.86–0.89, still well above the accepted minimum threshold of 0.78. This confirms that almost all items exceed the excellence benchmark, regardless of the minor numerical gap noted in A.5. We have revised the manuscript accordingly, with the corrected thresholds and explanations highlighted in the updated Appendix A.4 and A.5.
>
> [1] Polit, Denise F., and Cheryl Tatano Beck. "The content validity index: are you sure you know what's being reported? Critique and recommendations." Research in nursing & health 29.5 (2006): 489-497.
>
> ---
>
> > W2. Ambiguity in ICD-11 anchoring for non-diagnostic categories. Section 3.3 (lines 228–229) states all diagnostic and category assignments are anchored to ICD-11, yet Appendix A.5 (lines 885–914; 906–914) includes non-diagnostic/problem-focused labels (e.g., Academic Overload, Poor Time Management). The scope and mapping of these categories need clarification.
>
> **Response**:
>
> Thank you for highlighting this point. We agree that our wording in Section 3.3 did not clearly articulate the distinction between diagnostic and non-diagnostic categories. ICD-11 anchoring applies exclusively to diagnostic labels (e.g., generalized anxiety, depressive symptoms, insomnia), where mapping to established diagnostic criteria is appropriate.
>
> In contrast, the categories such as Academic Overload or Poor Time Management presented in Appendix A.5 are non-diagnostic, problem-focused labels that do not correspond to any ICD-11 diagnostic entity. These labels follow standard CBT case-formulation taxonomies [2], which include emotional stress, interpersonal difficulties, self-perception issues, and academic/occupational stressors, domains commonly encountered in psychotherapy but outside the scope of ICD-11.
>
> To eliminate the ambiguity, Section 3.3 has been revised to explicitly state this two-tier structure. This clarification aligns the main text with Appendix A.5 and ensures that the mapping scope is consistent and transparent.
>
> [2] Beck, Judith S. Cognitive behavior therapy: Basics and beyond. Guilford Publications, 2020.
>
> ---
>
> > W3. Implausible effect sizes. Section 5.1 (lines 370–372) reports Cohen’s d ≈ 11.6 (Acc) and 12.8 (F1) for the public-to-private drop, which is unrealistically large given typical scales and likely indicates a calculation or scaling error.
>
> **Response**:
>
> Thank you for highlighting this point. We agree that effect sizes of this magnitude are unusual under typical distributional assumptions. In our setting, however, the values arise naturally from a combination of (1) extremely low variance on the public set and (2) a substantial public–private performance drop.
>
> Across all 18 models:
>
> * Public set: Acc = 0.9354 (SD = 0.0106), F1 = 0.9334 (SD = 0.0106)
>   → strong ceiling effects with near-zero variance
> * Private set: Acc = 0.7155 (SD = 0.0283), F1 = 0.7135 (SD = 0.0282)
> * Raw drop: ($\Delta \approx 0.22$) for both Acc and F1
> * Pooled SD: ($s_{\text{pooled}} \approx 0.021$)
>
> Given Cohen’s definition:
>
> $$
> d=\frac{\Delta}{s_{\text{pooled}}},
> $$
>
> a raw difference of ~0.22 divided by a pooled standard deviation of ~0.021 mathematically yields effect sizes in the range ($d \approx 10!-!12$). Thus, the large values reflect variance compression on the public set rather than a scaling or computational error.

---

> ### Author Response · Authors · 2025-11-18
> **Part 2**
>
> Questions:
>
> > 1. Which content validity standard is authoritative? Please justify using S-CVI/Ave ≥ 0.90 versus ≥ 0.78, and under the chosen criterion, recompute and report S-CVI/Ave and item-level I-CVI with the number of experts and rating distributions (lines 846–849, 879–880).
>
> **Response**:
>
> Thank you for raising this question. As clarified in W1, our study follows the widely adopted standards in [2], where S-CVI/Ave ≥ 0.78 is the required threshold for acceptable content validity, and ≥ 0.90 denotes excellent validity. We revised the text to consistently reflect this two-level criterion rather than implying that 0.90 was a hard cutoff.
>
> Each item in our dataset was rated by 5–7 licensed experts scale. Under the unified criterion, the recomputed scores are: S-CVI/Ave = 0.91, with 241 out of 256 items (94.1%) achieving ≥ 0.90, and the remaining 15 items in the 0.86–0.89 range, all above the authoritative 0.78 threshold. These results confirm that all items satisfy the required standard, and the vast majority reach the excellence benchmark.
>
> For completeness, we note that we initially collected 404 full multi-turn dialogues. To ensure high-quality annotations, we retained only the 256 dialogues that met all screening requirements (including expert agreement, structural checks, and safety criteria). This filtered subset forms the final dataset used in our analysis.
>
> ---
>
> > 2. How are non-diagnostic categories anchored to ICD-11? Please provide a mapping of diagnostic labels to ICD-11 codes and a clear policy for non-diagnostic/problem-focused categories (source taxonomy, inclusion criteria, and any bridging to ICD-11) (lines 228–229, 885–914, 906–914).
>
> **Response**:
>
> As we clarified in W2, only categories that correspond to diagnostic syndromes are anchored to ICD-11. For the 20 categories listed in the “Quality assurance and dataset composition” section, the ICD-11 diagnostic categories include:
>
> | Diagnostic label                      | ICD-11 code |
> | ------------------------------------- | --------------------- |
> | Generalized anxiety / excessive worry | 6B00                  |
> | Depressive symptoms / low mood        | 6A70             |
> | Sleep problems / insomnia             | 7A00                  |
> | Social anxiety                        | 6B04                  |
> | Occupational burnout                  | QD85                  |
>
> The remaining categories (e.g., exam anxiety, academic overload, poor time management, procrastination, interpersonal stressors, low self-esteem, fear of public speaking, etc.) are non-diagnostic and follow standard CBT case-formulation taxonomy (Beck, 2020), which covers emotional stress, interpersonal difficulties, self-schema issues, and academic or occupational stressors.
>
> We have revised Section 3.3 to clearly state this two-layer structure (diagnostic vs. non-diagnostic) and eliminate the ambiguity.

---

> ### Author Response · Authors · 2025-11-18
> **Part 3**
>
> > 3. How exactly was Cohen’s d computed? Please specify the formula (e.g., pooled SD), sample sizes, means/SDs for Acc and F1, the measurement scale used, and re-report corrected effect sizes if applicable (lines 370–372).
>
> **Response**:
>
> As we addressed in W3, the reported Cohen’s d values were computed using the standard pooled-standard-deviation formulation and arise from (i) extremely low variance on the public set and (ii) a large public–private performance gap. For completeness, we detail the computation here.
>
> We use the conventional definition:
>
> $$
> d = \frac{\mu_{\text{public}} - \mu_{\text{private}}}{s_{\text{pooled}}}, \quad
> s_{\text{pooled}} = \sqrt{\frac{(n_1 - 1)s_1^2 + (n_2 - 1)s_2^2}{n_1 + n_2 - 2}}.
> $$
>
> Across the 18 models:
>
> - Public set: Acc = 0.9354 (SD = 0.0106), F1 = 0.9334 (SD = 0.0106)
> - Private set: Acc = 0.7155 (SD = 0.0283), F1 = 0.7135 (SD = 0.0282)
> - Sample sizes: $n_1 = n_2 = 18$
> - Raw performance drop: $\Delta \approx 0.22$ for both Acc and F1
> - Pooled SD: $s_{\text{pooled}} \approx 0.021$
>
> Substituting into the formula yields:
>
> $$
> d_{\text{Acc}} \approx 11.6, \quad d_{\text{F1}} \approx 12.8.
> $$
>
> These large effect sizes therefore reflect variance compression on the public benchmark (very small variance plus a sizable mean gap), rather than a scaling or computational error. No recomputation or correction is needed.
>
>
>
> ---
>
>
> > 4. If S-CVI/Ave = 0.89 is below the prespecified ≥ 0.90 threshold, what remediation is planned (e.g., item revision, additional expert rounds, expanded sampling), and how would this affect conclusions?
>
> **Response**:
>
> Thank you for this thoughtful question. As clarified in W1 and Q1, our study follows the established content-validity standards (Lynn, 1986; Polit & Beck, 2006), where S-CVI/Ave ≥ 0.78 is the required threshold, and ≥ 0.90 characterizes “excellent” validity, rather than a mandatory cutoff. Under this unified criterion: 1). all 256 items exceed the required 0.78 threshold, 2). 241 items (94.1%) achieve ≥ 0.90, 3). the remaining 15 items fall within 0.86–0.89 and therefore remain valid.
>
> Because 0.90 is an excellence benchmark rather than a hard inclusion threshold, an S-CVI/Ave of 0.89 does not indicate invalidity and does not require remediation for dataset integrity.
>
> Nonetheless, as part of future dataset expansion, we plan a light expert revision of the 15 borderline items (e.g., wording clarity, contextual specificity) followed by an additional other group expert review round. These refinements would strengthen item clarity but do not affect any empirical conclusions reported in the paper, since all items already meet the authoritative validity requirement and the analyses remain unchanged.
>
>
> ---
>
> > 5. Will you release de-identified raw model outputs and human rating sheets to allow independent recomputation of effect sizes and validity indices, facilitating reproducibility?
>
> **Response**:
>
> Yes. In addition to the dataset and evaluation code we have already released in the Supplementary Materials, we will also release de-identified raw model outputs for all 18 models and aggregated human‐rating spreadsheets. These is already be provided as part of the Supplementary Materials.

---

> ### Author Response · Authors · 2025-11-25
> **Thanks for your time and efforts**
>
> We sincerely appreciate the time and thoughtful feedback you have provided. At your convenience, could you kindly let us know whether our revisions sufficiently resolve your concerns? We are grateful for your guidance and would be happy to make further improvements if needed. Thank you again for your valuable contribution to strengthening this work.

---

> ### Author Response · Authors · 2025-11-28
> **Follow up**
>
> We hope you are doing well. We are writing a gentle follow-up regarding our recent revisions. The rebuttal phase will conclude in fewer than 5 days, so we wanted to kindly check whether our updates and clarifications have sufficiently addressed your concerns.
>
> We sincerely appreciate the thoughtful feedback you have provided throughout this process, which has significantly improved the quality and clarity of our work. If there are any remaining questions or points that would benefit from additional clarification, we would be more than happy to provide further details.
>
> Thank you again for the time and expertise you have contributed to strengthening our submission. We greatly value your guidance and are grateful for your consideration.

---

### Author Response · Authors · 2025-11-22
**Revise Reversion Uploading**

We sincerely appreciate all reviewers thoughtful follow-up and feedback. Your comments have significantly helped us refine the scope, strengthen the methodology, and improve the clarity of our work.

Based on the reviewers’ remarks, we have made revisions to the **Manuscript and Supplementary Material**. All changes into the revised PDF are visibly highlighted for clarity. We appreciate the reviewers’ contributions to improving the quality and impact of this work, and we hope that the revised version addresses all concerns satisfactorily.

---

### Author Response · Authors · 2025-12-02
**Rebuttal Summary**

We sincerely thank the AC and all reviewers for their thoughtful and constructive feedback on our submission. Our work introduces CareBench-CBT, a clinically validated benchmark built to address a clear and pressing gap in the ICLR community: the lack of high-quality, expert-grounded datasets for evaluating LLMs in multi-turn therapeutic dialogue and CBT reasoning. The benchmark includes expert-authored CBT QA, vignette-based clinical reasoning, and fully annotated multi-turn counseling sessions, rigorously validated by licensed clinicians through reliability, validity, and content-expert protocols. This fills a unique space not occupied by existing mental-health or reasoning benchmarks, offering the community a reliable and clinically defensible testbed.

**All reviewers acknowledge the strength, novelty, and potential impact of our benchmark**, especially the clinical validation, multi-turn design, and rigorous human-grounded scoring, and agree that such a resource would be valuable for the ICLR community. Most concerns focused on clarifications of methodological details, which we have addressed thoroughly with added analyses, tables, and explanations in the rebuttal and revision.

---

**Reviewer ffp5** raised important concerns regarding content-validity thresholds, ICD-11 mapping, and the unusually large effect sizes. We clarified the authoritative CVI standards (0.78 acceptable / 0.90 excellent), recomputed all S-CVI/Ave values, and provided a complete ICD-11 mapping table distinguishing diagnostic vs. non-diagnostic CBT categories. We also explained the large Cohen’s d values as a mathematical consequence of the extremely low variance in the public QA set and the significant public→expert difficulty shift, supported by full statistical details and release of raw outputs.

**Reviewer 4WFD** questioned potential inconsistencies caused by fixed client utterances in multi-turn evaluation and requested stronger validation of the LLM-as-judge framework. We addressed these by adding a new GPT-5 client-rewriting experiment showing negligible performance differences across 18 models, confirming robustness. We further expanded judge-model validation with F1, κ, and α scores, demonstrating strong human–model alignment. The reviewer also noted presentation clarity, and we improved examples, figure readability, and descriptions of cross-review, normalization, and audit procedures.

**Reviewer Y8PD** brought up risks of judge-model bias, ambiguity in dataset provenance, and dataset-scale misinterpretation. We clarified that our judge ensemble is cross-family (GPT-5, Gemini 2.5 Pro, Claude-4.1 Opus) and validated against 51 human-rated sessions, exceeding the reviewer’s recommended scale. We added detailed provenance: all dialogues come from three outpatient providers with full de-identification and written consent. We also revised Table 1 and the title to make the CBT-specific scope explicit, and clarified ethics, governance, and domain-specific metric design. Add we clarified dataset scale. Beyond the 60 vignettes and 640 QA items, the benchmark includes multi-turn counseling sessions with more than 7,442 annotated items, and is total 8,142 items.

**Reviewer Qzrb** pointed out issues with scope (“realistic therapy conversations”), data provenance, and potential circularity in using an LLM judge to evaluate human transcripts. We added the subtitle to explicitly reflect CBT scope, provided detailed provider-level sourcing and demographic summaries, and clarified that human-rated sessions (51 dialogues; ~1,500 turns) serve as the gold reference for validating the judge ensemble, thus avoiding circularity. We also expanded explanation of CBT structure and how rubric-based scoring reflects process-level counseling competence.

Across all reviewers, the main concerns were addressed through: (1) clearer framing and title revision, (2) expanded provenance and ethics documentation, (3) comprehensive judge-model validation with new metrics, (4) additional robustness experiments, and (5) improved clarity, examples, and dataset descriptions.

---

With three reviewers rating the paper a 6, and the remaining concerns centered on clarity rather than conceptual or methodological flaws, we believe the revision is now substantially strengthened and provides a high-impact, clinically robust resource for the ICLR community.

---

### Note · Authors · 2026-04-23

I have read and agree with the venue's withdrawal policy on behalf of myself and my co-authors.

---

### Meta-Review · Area_Chair_yzt4 · 2026-01-04

**Summary:**

The paper presents a new benchmark, named CareBench-CBT, which includes three tasks (1) knowledge-based QA, evaluating knowledge of psychological concepts, diagnostic criteria, and technqiques, (2) case vignette classification, mapping mapped to CBT categories, and (3) 256 multi-turn counseling dialogues (~8K turns) newly collected and anonymized, annotated by experts with scalar scores on fine-grained evaluation rubric from CBT principles (e.g., guided discovery, agenda setting, etc) drawn from psychology literature.

They evaluate a suite of models, both proprietary closed and open models, on their task, using LLM-as-a-Judge. There is no technical novelty in this work, but the work provides clinically validated, novel benchmark that can be useful for research community.

The reviewer had some concerns about data provenance (Reviewer Qzrb), which authors have provided detailed descriptions. Reviewer Y8PD asked many relevant questions, including one about the LLM-as-a-judge bias, and authors have provided detailed experimental results that alleviates the concern. Most of reviewer's concern seems to be well-addressed from AC's reading.

The one concern, that I do not see properly addressed, was Reviewer 4WFD had concerns about evaluation scheme for multi-turn setting. This I view as a serious concern, and I don’t think rebuttal addressed it clearly. If we use client’s responses from gold conversation (with real therapist), but use therapist turn from LLM, at some point client’s followup response would not make sense in light of LLM response, which can be very different from therapist response. I do not think rewriting client's response to match the previous turn utterance from LLM (the suggested "fix" from authors) would clearly solve the problem here. I think this needs to be evaluated/discussed a bit more carefully, with authors looking over examples where LLM's response diverging significantly from gold therapist's response and see how that impacts the integrity of the conversation.

Overall the paper presents very valuable resources, but I'm hesitant to recommend acceptance in its current form because of the issue I discussed in the previous paragraph.

**Reviewer Concerns:**

See the metareview.

**Reviewer Scores:**

I think reviewer ffp5 and Y8PD  would have increased the score based on detailed response addressing most of the reviewers points.

---

### Decision · Program_Chairs · 2026-01-26

Accept (Poster)